# EFFICIENT LOW-RANK AND SPARSE APPROXIMATION AND ADAPTATION FOR LARGE LANGUAGE MODELS

## ABSTRACT

Large Language Models (LLMs) have recently emerged as a significant advancement in natural language processing; however, their large scale and computational complexity make deployment a challenge. Model pruning has emerged as a post-training strategy to reduce LLMs' memory and computation needs. Despite notable progress, these techniques show a reduction in performance and necessitate post-pruning for recovery. To address these problems, we introduce **ELSA**, a novel method combining pruning and low-rank decomposition for better compression and recovery. We first use an alternating projections method to decompose the weight matrices into sparse matrices and low-rank matrices, which is validated from both theoretical and empirical perspectives; then we freeze the sparse matrices and update the low-rank matrices to efficiently recover the performance. To demonstrate the effectiveness and efficiency of the method, we conduct experiments on various language tasks (seven zero-shot tasks and language modeling) and models from different families (LLaMA, OPT, and Qwen) and different scales. The experiments show that the method outperforms state-of-the-art pruning methods and has comparable inference efficiency.

## 1 INTRODUCTION

Large Language Models (LLMs) have recently emerged as a significant advancement in natural language processing, delivering impressive outcomes across a range of tasks (Zhao et al., 2023). Nevertheless, their achievements are largely due to increased scale and computational complexity, leading to challenges in their storage and time efficiency. Post-training model compression (Zhu et al., 2023; Gholami et al., 2022; Hoefler et al., 2021) has aroused great interest as it can reduce the memory and computational demands of these models.

Pruning and low-rank approximation are both effective techniques for LLM compression. Pruning sets several elements in the weight matrices of the model to zero. It can be categorized into unstructured and structured pruning based on the sparse pattern. Low-rank approximation is to approximate the weight matrices using low-rank matrices, which can be regarded as a kind of structured pruning. Traditional pruning techniques (Ma et al., 2023; Huang et al., 2020; Han et al., 2015) often require a post-pruning retraining to recover performance. However, this is challenging in LLMs due to the model size. To address this limitation, post-training methods without retraining, such as Wanda (Sun et al., 2023) and SparseGPT (Frantar & Alistarh, 2023), are proposed. These methods can prune LLMs in a single forward pass without any gradient computation or recovery fine-tuning.

Although previous pruning methods can significantly accelerate end-to-end LLM inference, their performance drops a lot, and post-pruning retraining is needed to recover the performance. Therefore, efficiently finetuning the compressed model is an important topic in model compression. LLM-Pruner (Ma et al., 2023) is the first to propose using Low Rank Adaptation (LoRA) (Hu et al., 2022) to finetune part of the weights. LoSA (Huang et al., 2025) modified LoRA by adding a mask to low-rank matrices in order to merge low-rank adapters into the sparse weights. Although these methods can finetune the compressed model with low resources, they will either introduce additional parameters and destroy the sparseness or converge slowly and take more time to finetune.

To address these challenges, we present ELSA (**E**fficient **L**ow-rank and **S**parse **A**pproximation and **A**daptation), a novel method that compresses the model into a combination of a low-rank matrix and a sparse matrix, allowing for efficient fine-tuning of the compressed model. We first use an alternating

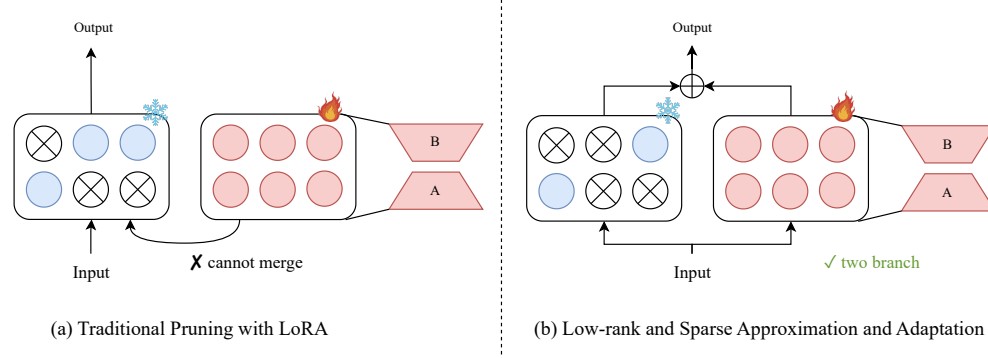

Figure 1: Difference between sparse only compression and ELSA . Traditional pruning methods transform model weight to a sparse matrix and introduce additional parameters during fine-tuning stage. ELSA decomposes the weight into a low-rank matrix and a sparse matrix, and transform the layer into a two branch layer. Fine-tuning the low-rank matrix will not break the sparsification and reduce inference latency.

projections method to decompose the weight matrices into sparse matrices and low-rank matrices. After the decomposition, we freeze the sparse matrices and update the low-rank matrices to efficiently recover the performance. ELSA supports unstructured and structured sparse patterns and can achieve the same theoretical computational complexity as pruning.

We compare ELSA with state-of-the-art pruning, low-rank approximation, and recovery fine-tuning with LoRA. To demonstrate the effectiveness and efficiency of ELSA , we conduct experiments on various language tasks (seven zero-shot tasks and language modeling), and models from different families (LLaMA (Touvron et al., 2023; Dubey et al., 2024), OPT (Zhang et al., 2022), and Qwen (Team, 2024)) and different scales. We also evaluate the compression, finetuning, and inference efficiency of this decomposition form in both CPU and GPU. The results show that ELSA can outperform state-of-the-art methods before and after recovery finetuning in different compression settings and can achieve better speedup on both CPU and GPU. The contributions of this work are summarized as follows:

- *Novel Decomposition Method*: ELSA proposes to approximate the weight matrices with the combination of sparse matrices and low-rank matrices using an alternating projections method with improved sub-problem solver to minimize approximation error. This algorithm is supported by theoretical proof and empirical experiments.

- *Efficient Recovery Finetuning*: The compressed format of ELSA is naturally compatible with LoRA finetuning. Compared with other recovery finetuning strategies, ELSA updates the low-rank matrices and freezing sparse matrices, which can reduce memory usage, accelerate the training speed, and enhance the performance.

- *Layer-wise Compression Rate*: ELSA replaces uniform compression rate by layer-wise rate, which is computed by activation similarity and outliers proportion. This strategy can identify the importance of different layers and allocate appropriate rates among them, leading to better performance.

- *Comprehensive Experimental Evaluation*: We conduct comprehensive experiments to evaluate ELSA across different settings and models from different families and scales. We also test the inference speed of the compressed models on CPU and GPU based on DeepSparse (Magic, 2021) and nm-vllm (Magic, 2024). The experiments show that ELSA outperforms state-of-the-art pruning and low-rank approximation methods and has comparable inference efficiency.

## 2 EFFICIENT LOW-RANK AND SPARSE APPROXIMATION AND ADAPTATION

The key insight of ELSA is that since both sparsity and low-rank decomposition are effective ways to reduce network parameters, the combination of them can store the parameters in different ways and

enhance the performance of the compressed model. Moreover, since LoRA has become a popular method to fine-tune LLMs, the format of this compressed form is naturally compatible with LoRA finetuning. The decomposition format of ELSA is $W = L + S$, where $L$ is a low-rank matrix and $S$ is a sparse matrix. ELSA can be summarized in three parts: matrix decomposition, efficient recovery fine-tuning, and layer-wise compression rate.

## 2.1 MATRIX DECOMPOSITION

Following other layer-wise compression methods, we formulate the decomposition as an optimization problem.

$$\min_{L,S} \|X(W - L - S)^T\|_F, \text{ s.t. } \text{rank}(L) \leq r, \|S\|_0 \leq s \tag{1}$$

### 2.1.1 ALTERNATIVE PROJECTIONS METHOD FOR UNSTRUCTURED SPARSE PATTERN

Without any structural limitation for $S$, this problem is similar to the Robust PCA (Candès et al., 2011). To solve it, we use the alternating projections method (Netrapalli et al., 2014), which alternately solves two sub-problems: fix $L$ to optimize $S$ and fix $S$ to optimize $L$.

Truncated SVD on $L$ is capable of minimizing the approximation error when fixing $S$.

**Theorem 2.1.** *Denote the eigenvalue decomposition of $X^T X$ as $Q\Lambda Q^T$, and the singular value decomposition of $(W - S)Q\Lambda^{1/2}$ as $U\Sigma V^T$. Let $U_r, \Sigma_r, V_r = U[:, :r], \Sigma[:r, :r], V[:, :r]$, then $L = U_r\Sigma_r V_r^T \Lambda^{-1/2} Q^T$ can minimize Problem 1 when $S$ is fixed.*

*Proof.* To simplify the writing, we exclude $S$ in the equations. Problem 1 can be transformed as follows.

$$\begin{aligned}
\min \|X(W - L)^T\|_F^2 &\Leftrightarrow \min \text{tr}((W - L)X^T X(W - L)^T) \\
&\Leftrightarrow \min \text{tr}(((W - L)Q\Lambda Q^T(W - L)^T) \\
&\Leftrightarrow \min \|\Lambda^{1/2}Q^T(W - L)^T)\|_F^2 \\
&\Leftrightarrow \min \|WQ\Lambda^{1/2} - LQ\Lambda^{1/2}\|_F^2
\end{aligned}$$

This is a low-rank approximation problem. According to the Eckart–Young–Mirsky theorem, $LQ\Lambda^{1/2} = U_r\Sigma_r V_r^T$ can minimize the problem, where $U_r, \Sigma_r, V_r$ are the top-$r$ singular vectors and values of $WQ\Lambda^{1/2}$. Consequently, $L = U_r\Sigma_r V_r^T \Lambda^{-1/2} Q^T$ minimizes Problem 1 when fixing $S$. ∎

Fixing $L$ and Optimizing $S$ is a pruning problem, which can be solved by any existing pruning methods. In order to reduce computational overhead, we use a fast and efficient method, Wanda (Sun et al., 2023), to solve this problem. Formally, let $D = \textbf{Diag}(\sqrt{X^T X})$, Wanda calculates $S = \textbf{Truncate}(WD, s)D^{-1}$, which keeps $s$ largest elements in magnitude in $WD$ and sets the rest to zero.

When alternatively solving these two sub-problems, this truncated method does not perform well enough. The main reason is that the truncated method easily drops to a local minimum. **Truncate**() will generate a lot of irregular zeros, which is challenging for low-rank approximation. Therefore, we introduce a shrink function to solve $S$. We use **Shrink**() to replace **Truncate**(), which is defined as:

$$\textbf{Shrink}(W, s, \tau)_{ij} = \textbf{Sign}(W_{ij}) \cdot \begin{cases} 0, & |W_{ij}| \leq \eta \\ |W_{ij}| - \eta\tau, & |W_{ij}| \geq \eta \end{cases}, \eta = \textbf{Top}_s(|W_{ij}|), \tau = \exp(-T)$$

Here, $T$ denotes the iteration step, and $\tau$ reduces as the iteration progresses.

To achieve the target compression rate $p$, $r$, and $s$ should satisfy the equation: $(d_{\text{in}} + d_{\text{out}})r + s = (1 - p)d_{\text{in}}d_{\text{out}}$. Following OATS (Zhang & Papyan, 2024), we introduce a hyperparameter rank

**Algorithm 1** Unstructured ELSA

**Inputs:**
    $W \in \mathbb{R}^{d_{\text{out}} \times d_{\text{in}}}$: Weight matrix
    $X \in \mathbb{R}^{l \times d_{\text{in}}}$: Input matrix
    $N$: Iterations
    $p$: Compression rate
    $k$: Rank ratio

Calculate $r$ and $s$ using Equation 2.
$D = \mathbf{Diag}(\sqrt{X^T X})$
$Q \Lambda Q^T = \mathbf{Eig}(X^T X)$
$L = 0$
**for** $t = 1$ to $N$ **do**
    $S = \mathbf{Shrink}(WD, s, \exp(-t))D^{-1}$
    $U\Sigma V^T = \mathbf{SVD}((W - S)Q\Lambda^{1/2})$
    $L = U_r \Sigma_r V_r^T \Lambda^{-1/2} Q^T$
**end for**
**return**: $S, L$

**Algorithm 2** Structured ELSA

**Inputs:**
    $W \in \mathbb{R}^{d_{\text{out}} \times d_{\text{in}}}$: Weight matrix
    $X \in \mathbb{R}^{l \times d_{\text{in}}}$: Input matrix
    $p$: Compression rate
    $k$: Rank ratio

Calculate $r$ and $d_S$ using Equation 3.
$Q \Lambda Q^T = \mathbf{Eig}(X^T X)$
Calculate $\|W_i Q_i \Lambda^{1/2}\|_F$ and choose the top-$d_S$ smallest columns $i$
$S = W[:, -i], W_{-S} = W[:, i]^a$
$\tilde{Q} \tilde{\Lambda} \tilde{Q}^T = \mathbf{Eig}(X^T X[i, i])$
$U\Sigma V^T = \mathbf{SVD}(W_{-S} \tilde{Q} \tilde{\Lambda}^{1/2})$
$L = U_r \Sigma_r V_r^T \tilde{\Lambda}^{-1/2} \tilde{Q}^T$
**return**: $S, L$

---
$^a -i$ represents the rest indices.

ratio $k$ to represent the proportion of nonzero parameters in the low-rank branch, which means $k = (d_{\text{in}} + d_{\text{out}})r/(1 - p)d_{\text{in}}d_{\text{out}}$. Therefore, given the compression rate $p$ and the rank ratio $k$, $r$ and $s$ are calculated by

$$r = \left\lfloor \frac{kp d_{\text{in}} d_{\text{out}}}{(d_{\text{in}} + d_{\text{out}})} \right\rfloor, s = \lfloor (1 - k)p d_{\text{in}} d_{\text{out}} \rfloor \tag{2}$$

The entire alternating projections method is shown in Algorithm 1. In practice, running standard SVD takes much time. Since our method needs several iterations, we use randomized SVD (Halko et al., 2011) to get the truncated singular vectors and values instead, which can get a close solution to the matrix approximation problem and run much faster.

### 2.1.2 EXPANDING TO STRUCTURED DECOMPOSITION

When expanding to Structured Decomposition, there are a few subtle differences from the afore-mentioned method. In the context of structured decomposition, to fully employ the parameters, both $S$ and $L$ exhibit structured sparsity, implying that they compute input channels separately. To calculate $XW^T$, the input $X \in \mathbb{R}^{l \times d_{\text{in}}}$ is rearranged into two segments: $X_S \in \mathbb{R}^{l \times d_S}$ and $X_L \in \mathbb{R}^{l \times d_L}$. These segments are then multiplied by $S \in \mathbb{R}^{d_{\text{out}} \times d_S}$ and $L \in \mathbb{R}^{d_{\text{out}} \times d_L}$, respectively, and subsequently combined as follows: $XW^T = X_S S^T + X_L L^T$.

To achieve the structured decomposition, we initially select certain columns from $W$ to create $S$ and then apply truncated SVD to the rest of the matrix $W_{-S}$. Since we do low-rank approximation on $W_{-S} Q_{-S} \Lambda^{1/2}$, to reduce the approximation error, we heuristically choose the columns that minimize $\|W_{-S} Q_{-S} \Lambda^{1/2}\|_F$ to form $W_{-S}$. We choose the columns with the smallest values of $\|W_i Q_i \Lambda^{1/2}\|_F$ based on the observation that $\|W_{-S} Q_{-S} \Lambda^{1/2}\|_F = \|\sum W_i Q_i \Lambda^{1/2}\|_F \leq \sum \|W_i Q_i \Lambda^{1/2}\|_F$. Notice that $Q_{-S} \Lambda^{1/2}$ is not of full rank, running SVD on $W_{-S} Q_{-S} \Lambda^{1/2}$ will yield a wrong $L$. Therefore, another eigenvalue decomposition should be applied on $X_{-S}^T X_{-S}$, followed by a truncated SVD.

The allocation of parameters in the low-rank and sparse part is also different. Remind that rank ratio $k$ represents the proportion of nonzero parameters in the low-rank branch. In structured decomposition, given the compression rate $p$ and rank ratio $k$, $r$ and $s$ are calculated by

$$d_S = \lfloor (1 - k)p d_{\text{in}} \rfloor, \ r = \left\lfloor \frac{kp d_{\text{in}} d_{\text{out}}}{(d_{\text{in}} + d_{\text{out}} - d_S)} \right\rfloor \tag{3}$$

The entire decomposition algorithm is shown in Algorithm 2.

### 2.1.3 COMPUTATIONAL COMPLEXITY ANALYSIS

Assume the weight matrix $\boldsymbol{W} \in \mathbb{R}^{d_{\text{out}} \times d_{\text{in}}}$ is decomposed into $\boldsymbol{S} \in \mathbb{R}^{d_{\text{out}} \times d_{\text{in}}}$, $\boldsymbol{A} \in \mathbb{R}^{r \times d_{\text{in}}}$, and $\boldsymbol{B} \in \mathbb{R}^{d_{\text{out}} \times r}$. Given input $\boldsymbol{X} \in \mathbb{R}^{l \times d_{\text{in}}}$, ELSA calculates $\boldsymbol{X}\boldsymbol{W}^T$ by $\boldsymbol{X}\boldsymbol{S}^T + (\boldsymbol{X}\boldsymbol{A}^T)\boldsymbol{B}^T$. The computational complexity of dense matrix multiplication for two matrices of sizes $m \times r$ and $r \times n$ is $O(mnr)$. The computational complexity of sparse matrix multiplication for a dense matrix of size $m \times r$ and a sparse matrix of size $r \times n$ with $s$ non-zero elements is $O(ms)$. Therefore, the overall computational complexity is $O(ls + ld_{\text{in}}r + ld_{\text{out}}r)$. Since we have $(d_{\text{in}} + d_{\text{out}})r + s = (1-p)d_{\text{in}}d_{\text{out}}$ in the definition of $p$, the computational complexity is equal to $O((1-p)ld_{\text{in}}d_{\text{out}})$, which is $1-p$ of the original layer. Theoretically, this compression format has the same computational complexity as sparsity. In practice, accelerating the low-rank branch is straightforward, and increasing the sparsity of the sparse branch further boosts its acceleration, resulting in superior overall acceleration.

## 2.2 COMPATIBLE WITH LoRA FINETUNING

Directly running LoRA finetuning on unstructured or semi-structured sparse models will introduce more parameters or destroy the sparsity. To narrow this gap, LoSA introduced masks in the LoRA finetuning, which can preserve the sparsity. Our method solves this problem from another perspective. The low-rank branch in the compressed model can be directly used to finetune without introducing more parameters or destroying the sparsity. For each compressed layer $\boldsymbol{W} = \boldsymbol{S} + \boldsymbol{L} = \boldsymbol{S} + \boldsymbol{A}\boldsymbol{B}$, the low-rank branch $\boldsymbol{A}, \boldsymbol{B}$ is set to the initialization of low-rank matrices, and LoRA directly updates $\boldsymbol{A}$ and $\boldsymbol{B}$. If the memory budget is given, we choose vectors with larger singular values for training, which are the left columns in $\boldsymbol{A}$ and top rows in $\boldsymbol{B}$.

Running on the low-rank branch has several advantages except for preserving the sparsity:

**Compared with mask based methods** Our method removes the requirement for masks and accelerates training. Moreover, when using masks, the gradient of the mask elements is zero and has to be estimated, usually via STE (Straight Through Estimator) (Bengio et al., 2013), which is hard to ensure accuracy. Consequently, removing the mask can accelerate training and convergence.

**Compared with standard LoRA** We initialize the low-rank matrices $\boldsymbol{A}$ and $\boldsymbol{B}$ using the singular vectors corresponding to the largest singular values. The trainable component $\boldsymbol{L} = \boldsymbol{A}\boldsymbol{B}$ encapsulates the critical directions of $\boldsymbol{W} - \boldsymbol{S}$, which can lead to faster and more effective convergence. In the standard LoRA configuration, $\boldsymbol{A}$ and $\boldsymbol{B}$ are initialized to Gaussian noise and zeros. PiSSA (Meng et al., 2024) has shown that in dense models, this zero initialization will potentially cause insignificant and random directional gradients during the initial finetuning stage, and potentially squander many gradient descent steps. ELSA transfers this method to compressed models and shares the same advantage.

**Compared with dense model LoRA** The frozen part $\boldsymbol{S}$ is a sparse matrix, which can reduce memory and accelerate computation. In LoRA finetuning, gradients are calculated by: $\frac{\partial l}{\partial \boldsymbol{X}} = \frac{\partial l}{\partial \boldsymbol{Y}}(\boldsymbol{S} + \boldsymbol{A}\boldsymbol{B}), \frac{\partial l}{\partial \boldsymbol{A}} = \boldsymbol{X}^T \frac{\partial l}{\partial \boldsymbol{Y}} \boldsymbol{B}^T, \frac{\partial l}{\partial \boldsymbol{B}} = \boldsymbol{A}^T \boldsymbol{X}^T \frac{\partial l}{\partial \boldsymbol{Y}}$. During backward propagation, $\boldsymbol{S}$ contributes solely to the calculations in $\frac{\partial l}{\partial \boldsymbol{X}}$. While in the forward pass, $\boldsymbol{S}$ is also employed. It is important to note that operations involving $\boldsymbol{S}$ are all matrix multiplications. There are existing techniques for speeding up sparse matrix multiplications. By employing these techniques, frozen sparse matrices can boost the speed of LoRA finetuning.

## 2.3 LAYER-WISE COMPRESSION RATE ALLOCATION

The importance of different layers in LLM varies. Given the overall compression rate of the model, the parameters should be allocated according to the importance of each layer – the more important the layer, the lower compression rate it has. We use softmax function to allocate the compression rate in order to keep the overall compression rate and the hyper-parameter can control the allocation, which is also discussed in MoDeGPT(Lin et al., 2024).

The allocation can be divided into two part: block-wise and layer-wise. We first use the similarity of activations, which is denoted as Block Influence introduced by ShortGPT (Men et al., 2024) $(1 - X_{\text{in}}^T X_{\text{out}} / \|X_{\text{in}}\|_2 \|X_{\text{out}}\|_2)$ to measure the importance of a block. Then we allocate compression

Table 1: Zero-shot accuracy and perplexity of ELSA and baselines on LLMs from different families under different compression setting. The data in a cell represents the performance without and with recovery fine-tuning respectively. OATS is only employed without recovery fine-tuning.

| Compression Rate & Format | Method | LLaMA-2-7B | | LLaMA-3-8B | | OPT-6.7B | | Qwen2.5-7B | |
|---|---|---|---|---|---|---|---|---|---|
| | | 0-shot (%)($\uparrow$) | PPL ($\downarrow$) | 0-shot (%)($\uparrow$) | PPL ($\downarrow$) | 0-shot (%)($\uparrow$) | PPL ($\downarrow$) | 0-shot (%)($\uparrow$) | PPL ($\downarrow$) |
| Dense | - | 66.40 | 5.42 | 70.00 | 6.13 | 58.16 | 10.86 | 70.32 | 6.85 |
| 50% unstructured | Wanda | 64.37 / 63.74 | 6.90 / 6.77 | 63.36 / 66.39 | 9.81 / 8.75 | 55.70 / 56.94 | 11.98 / 12.30 | 66.23 / **68.89** | 8.58 / 8.20 |
| | SparseGPT | 63.83 / 63.13 | 6.92 / 6.79 | 65.27 / 68.15 | 9.46 / 8.89 | **56.87** / 57.45 | 11.65 / 12.07 | 67.09 / 68.22 | 8.47 / 8.16 |
| | OATS | 63.61 / - | 6.77 / - | 65.07 / - | 9.23 / - | 56.36 / - | 12.06 / - | 67.79 / - | 8.85 / - |
| | ELSA | **64.74 / 65.10** | **6.57 / 6.49** | **66.39 / 68.50** | **8.75 / 8.31** | 56.74 / **57.93** | **11.48 / 12.01** | **68.26** / 68.77 | **8.14 / 8.02** |
| 2:4 semi-structured | Wanda | 51.40 / 59.52 | 12.12 / 8.37 | 50.99 / 61.06 | 24.36 / 12.34 | 51.66 / 53.19 | 15.99 / 14.48 | 60.20 / 66.68 | 15.03 / 10.05 |
| | SparseGPT | 53.71 / 59.72 | 10.79 / 8.18 | 54.98 / 61.09 | 16.84 / 11.60 | 53.71 / 55.45 | 14.13 / 13.93 | 62.94 / 67.20 | 11.36 / 9.62 |
| | OATS | 54.61 / - | 10.98 / - | 53.99 / - | 18.30 / - | 54.00 / - | 14.69 / - | 59.35 / - | 13.83 / - |
| | ELSA | **55.55 / 60.44** | **10.39 / 8.08** | **58.93 / 63.44** | **13.82 / 11.45** | **54.86 / 56.40** | **12.77 / 13.40** | **63.21 / 68.86** | **10.91 / 8.16** |
| 20% structured | SliceGPT | 56.54 | 15.70 | 55.71 | 36.99 | 55.53 | 13.98 | 61.47 | **11.39** |
| | SVD-LLM | 54.53 | 11.36 | 53.00 | 78.87 | 56.99 | 13.23 | 58.71 | 22.98 |
| | ELSA | **56.74** | **9.31** | **59.07** | **22.06** | **57.13** | **12.83** | **63.53** | 11.84 |

rate among the linear layers in a Transformer block. Drawing inspiration from OWL (Yin et al., 2023), we utilize the outliers' proportion[1] as a criterion for evaluating layer significance. We adjust OWL for our compression format accordingly: we first subtract the principal singular vectors and then assess the outlier proportion of the remaining matrix. Layers with a higher ratio of outliers are deemed more significant and therefore require a smaller compression ratio.

# 3 EXPERIMENT AND ANALYSIS

## 3.1 EXPERIMENT SETUP

**Baseline.** ELSA can be applied to three types of decomposition structures based on the sparse pattern: unstructured, semi-structured, and structured. For unstructured decomposition, we compare with Wanda, SparseGPT fine-tuned by LoRA strategy and OATS. For structured decomposition, we compare with SOTA structured pruning and low-rank decomposition methods: SliceGPT (Ashkboos et al., 2024) and SVD-LLM (Wang et al., 2024) without recovery fine-tuning.

**Models and Evaluation.** We evaluate ELSA on widely used large language models of various scales: LLaMA-2-7B/13B (Touvron et al., 2023), LLaMA-3-8B (Dubey et al., 2024), LLaMA-3.2-1B/3B, OPT-6.7B (Zhang et al., 2022), and Qwen2.5-7B(Team, 2024). We employ the LM Evaluation Harness (Gao et al., 2021) to evaluate zero-shot accuracy on seven tasks: ARC Challenge (Clark et al., 2018), ARC Easy (Clark et al., 2018), BoolQ (Clark et al., 2019), HellaSwag (Zellers et al., 2019), OpenbookQA (Mihaylov et al., 2018), PIQA (Bisk et al., 2020), and WinoGrande (Sakaguchi et al., 2021). We report the average accuracy in the main text, and detailed results are provided in Appendix A.3. We also evaluate the perplexity on the WikiText-2 (Merity et al., 2016) test set.

**Implementation Details.** For the compression task, consistent with previous methods, we use C4 (Raffel et al., 2020) as the calibration dataset. We sample 128 sequences of length 2048 from the calibration set. We evaluate three types of decomposition categorized by sparse pattern: 50% unstructured, 2:4 semi-structured, and 20% structured. The iteration number is set to 80, and rank rate $k$ is set to 0.1. For the recovery finetuning task, we randomly sampled a 10K subset from the Alpaca-GPT4 (Peng et al., 2023) as the training dataset. We use the AdamW optimizer (Loshchilov & Hutter, 2017) with a learning rate of 1e-4 and cosine scheduler. The average rank is set to 32. All experiments can be conducted on a 96GB H20 GPU. More implementation details are provided in Appendix A.1 and A.2.

## 3.2 COMPARISON WITH SOTA PRUNING AND LOW-RANK DECOMPOSITION METHODS

In this section, we compare ELSA with SOTA pruning and low-rank decomposition methods across 50% unstructured, 2:4 semi-structured, and 20% structured settings. We compare from three aspects: (1) performance on LLMs from different families; (2) performance on LLMs of different scales; (3) error and loss curve of decomposition and fine-tuning process.

---

[1]The proportion is calculated by $\frac{\sum_{i=1}^{d_{\text{out}}} \sum_{j=1}^{d_{\text{in}}} \mathbb{I}(A_{ij} > M \cdot \bar{A})}{d_{\text{in}} d_{\text{out}}}$, $\boldsymbol{A} = |\boldsymbol{W}\boldsymbol{D}|$, $\bar{A}$ is the mean of $\boldsymbol{A}$, $M$ is an integer.

Table 2: Zero-shot accuracy and perplexity of ELSA and baselines on LLMs of different scales under different compression setting. The data in a cell represents the performance without and with recovery fine-tuning respectively. SliceGPT is not implemented for LLaMA-3.2 models.

| Compression Rate & Format | Method | LLaMA-3.2-1B | | LLaMA-3.2-3B | | LLaMA-2-13B | |
|---|---|---|---|---|---|---|---|
| | | 0-shot (%)($\uparrow$) | PPL ($\downarrow$) | 0-shot (%)($\uparrow$) | PPL ($\downarrow$) | 0-shot (%)($\uparrow$) | PPL ($\downarrow$) |
| Dense | - | 56.61 | 9.75 | 62.89 | 7.81 | 69.25 | 4.88 |
| 50% unstructured | Wanda | 46.56 / 51.30 | 24.39 / 17.91 | 53.64 / 57.99 | 13.11 / 11.50 | 66.93 / 67.23 | 5.94 / 5.82 |
| | SparseGPT | 49.36 / 51.31 | 21.83 / 16.67 | 56.58 / 58.87 | 12.43 / 11.37 | 67.18 / 67.39 | 5.98 / 5.79 |
| | OATS | 49.52 / - | 18.34 / - | 57.10 / - | 12.01 / - | 67.26 / - | 5.90 / - |
| | ELSA | **49.60 / 52.04** | **17.44 / 16.65** | **57.24 / 59.05** | **11.87 / 11.21** | **67.41 / 67.58** | **5.86 / 5.64** |
| 2:4 semi-structured | Wanda | 40.53 / 46.13 | 80.09 / 30.53 | 43.67 / 52.39 | 33.57 / 17.23 | 59.86 / 62.74 | 9.13 / 6.99 |
| | SparseGPT | 43.98 / 48.67 | 36.76 / 24.55 | 46.27 / 53.41 | 23.04 / 15.38 | 61.90 / 64.08 | 8.53 / 6.86 |
| | OATS | 45.82 / - | 25.56 / - | 47.83 / - | 20.85 / - | 63.45 / - | 7.02 / - |
| | ELSA | **46.47 / 49.02** | **25.31 / 20.91** | **49.25 / 56.26** | **20.62 / 13.03** | **64.34 / 65.19** | **6.84 / 6.61** |
| 20% structured | SliceGPT | - | - | - | - | 61.55 | 8.96 |
| | SVD-LLM | 41.02 | 105.80 | 47.41 | 78.87 | 60.60 | 8.92 |
| | ELSA | **41.52** | **42.69** | **47.87** | **24.78** | **63.87** | **7.70** |

**Performance on LLMs from different families.** To validate the generalizability of ELSA across different LLMs, we compare ELSA with baseline methods on four different LLMs of similar scales from different families: LLaMA-2-7B, LLaMA-3-8B, OPT-6.7B, and Qwen2.5-7B under different compression settings. These models have different model architectures and training data. As Table 1 shows, ELSA consistently outperforms baselines on the chosen LLMs under different compression settings without recovery fine-tuning. With recovery fine-tuning, ELSA outperforms baselines in most cases and achieves similar results in the rest of the cases. We highlight that in the unstructured setting, our method can improve the performance while keeping the model size.

**Performance on LLMs of Different Scales.** To validate the generalizability of ELSA across different model scales, we also compare ELSA with baseline methods on LLMs of larger and smaller scales from the LLaMA family: LLaMA-3.2-1B, LLaMA-3.2-3B, and LLaMA-2-13B. As shown in Table 2, ELSA outperforms baseline methods in LLMs of different scales under different compression settings.

**Approximation Error Curve of Decomposition.**
Figure 6 illustrates the approximation error for different sub-problem solvers. We choose q_proj and down_proj in the 15th layers in LLaMA-2-7B for this experiment. The $\mathrm{Shrink}()$ function shows a lower approximation error and faster convergence, leading to superior compression performance. Furthermore, the curves of randomized SVD and full SVD are close, which indicates that using randomized SVD can yield a proper result with high speed. We also find that in most layers of the LLM, adopting sparsification is more effective than low-rank decomposition, and combining both results in smaller errors.

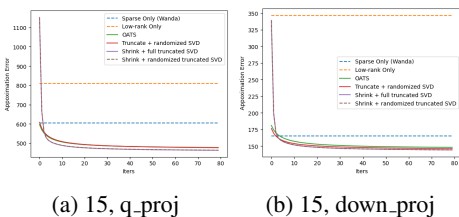

(a) 15, q_proj          (b) 15, down_proj

Figure 2: Approximation Error Curve of a single layer in LLaMA-2-7B.

## 3.3 EFFICIENCY OF ELSA

### 3.3.1 COMPRESSION AND FINETUNING SPEED

We report the compression speed and memory of different methods with LLaMA-2-7B in Table 3. Compared with Wanda and SparseGPT, OATS and ELSA need more time, mainly because of the matrix decomposition. Compared with OATS, ELSA uses randomized SVD to reduce the time and memory costs of SVD. In the recovery finetuning stage, the peak memory of LoRA and ELSA is the same, while ELSA saves 3.2% time (19.40 to 18.78 minutes).

Table 3: Compress speed and memory.

| Method | Time(min) | Peak Memory(GB) |
|---|---|---|
| **Wanda** | 2.67 | 9.63 |
| **SparseGPT** | 5.97 | 7.64 |
| **OATS** | 634.47 | 5.75 |
| **ELSA (unstructured)** | 21.68 | 10.55 |
| **SliceGPT** | 28.17 | 6.75 |
| **SVD-LLM** | 18.56 | 7.48 |
| **ELSA (structured)** | 19.21 | 7.60 |

Table 4: End-to-end inference speed of compressed LLaMA-2-7B on CPU and GPU.

| Device | Compression Rate | Dense | 30% | | | 40% | | | 50% | | |
|---|---|---|---|---|---|---|---|---|---|---|---|
| | | | Pruning+LoRA | Pruning | ELSA | Pruning+LoRA | Pruning | ELSA | Pruning+LoRA | Pruning | ELSA |
| CPU | Throughput (tokens/sec) | 3.73 | 4.15 | 4.35 | 4.78 | 4.86 | 5.23 | 5.96 | 7.00 | 7.19 | 8.12 |
| | Speedup | 1.00× | 1.11× | 1.17× | 1.28× | 1.30× | 1.40× | 1.60× | 1.88× | 1.93× | 2.18× |
| GPU | Throughput (tokens/sec) | 58.48 | 63.15 | 65.67 | 66.49 | 70.18 | 78.35 | 80.84 | 79.72 | 92.13 | 98.52 |
| | Speedup | 1.00× | 1.08× | 1.12× | 1.14× | 1.20× | 1.34× | 1.37× | 1.36× | 1.58× | 1.68× |

### 3.3.2 INFERENCE SPEEDUP ON HARDWARE

We test the inference speed of different types of decomposition on CPU and GPU, using LLaMA-2-7B as an example, respectively. Specifically, unstructured sparsification can be applied to CPU acceleration. Unstructured and 2:4 semi-structured sparsification can be applied to GPU acceleration.

For CPU acceleration, we utilize DeepSparse engine (Magic, 2021) to measure CPU acceleration of the unstructured sparse pattern. We run a LLaMA-2-7B model for a single batch of 2048 tokens on Intel(R) Xeon(R) Platinum 8480CL @ 2.0 GHz with 56 cores. For GPU acceleration, we utilize nm-vllm (Magic, 2024) engine to measure GPU acceleration of unstructured and 2:4 structured sparse patterns. We run a LLaMA-2-7B model for a single batch of 2048 tokens on an RTX 4090. The unstructured acceleration results are shown in Table 4, while 2:4 semi-structured will introduce 8% latency. Compared with the dense model, ELSA can achieve 1.28-2.18 × on CPU and 1.14-1.68 × speedup on GPU. On the other hand, applying LoRA fine-tuning after pruning introduces additional parameters, resulting in increased latency.

### 3.4 ABLATION STUDY

**Effect of Key Components in Decomposition.**
There are three key components in the decomposition: substitute $\mathrm{Truncated}()$ with $\mathrm{Shrink}()$, swap the scaling matrix $D$ for eigenvectors, and use a layer-wise compression. We evaluate the individual effects and the combined impact of these three components. Denote these three components as SHR, EIG, and LWC respectively. Table 5 shows the effect of these components. Removing one or all components will result in performance degradation. The results demonstrate that all three key components contribute to the final accuracy, and using all three strategies can achieve the best performance.

Table 5: Ablation experiment on key components.

| Method | 0-shot (%)(↑) | PPL (↓) |
|---|---|---|
| **ELSA** | **64.74** | **6.57** |
| w/o SHR | 64.31 | 6.79 |
| w/o EIG | 64.11 | 6.75 |
| w/o LWC | 64.40 | 6.62 |
| w/o SHR & EIG & LWC | 64.08 | 6.76 |

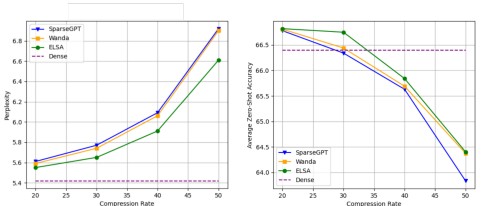
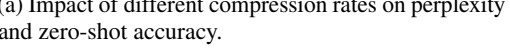
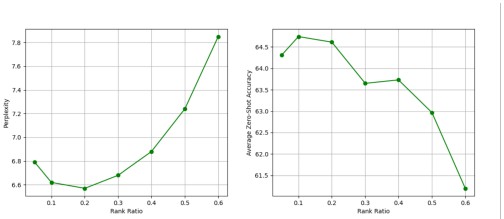

(a) Impact of different compression rates on perplexity and zero-shot accuracy.

(b) Impact of different rank ratios on perplexity and zero-shot accuracy.

Figure 3: Ablation experiment on compression rate and rank ratio.

**Impact of Different Compression Rates.** We evaluate ELSA and baseline methods at different compression rates on LLaMA-2-7B. As shown in Figure 3a, ELSA outperforms baseline methods under different compression rates. When the compression rate is lower than 30%, the compressed model outperforms the dense model on zero-shot tasks.

**Impact of Different Rank Ratios.** We evaluate ELSA under different rank ratios on LLaMA-2-7B. Rank ratios represent the proportion of low-rank matrix in the compressed model. It not only affects the performance but also the speed of performing randomized SVD, which is a time-consuming operation. As shown in Figure 3b, ELSA performs better with smaller rank ratios, and smaller rank ratios result in faster compression. Rank ratios between 0.1 and 0.2 can achieve the best performance.

## 4 Related Work

### 4.1 Model Compression

Model compression, as a post-training technique, has aroused great interest since it can reduce the memory and computational requirements of large language models. It usually includes distillation, quantization, pruning, and low-rank approximation. ELSA combines two of these techniques: pruning and low-rank approximation, to enhance the compressed model's performance. In particular, when the rank ratio is 0, ELSA executes pruning as Wanda (Sun et al., 2023), and when the rank ratio is 1, it executes low-rank approximation similar to SVD-LLM (Wang et al., 2024).

### 4.2 Low-rank Adaptation of Large Language Model

Low-rank adaptation (LoRA) has become a simple but effective technique for reducing memory usage while finetuning large language models (LLMs). LoRA reduces memory footprint by removing the need to save gradients and the associated optimizer states (e.g., the momentum and variance statistics in Adam) for the base model. Recently, many works studied LoRA and proposed variants of LoRA. Among these methods, PiSSA (Meng et al., 2024) proposed directly tuning the principal singular vectors and values of weight matrices, allowing faster convergence and enhanced performance. ELSA extends the idea of finetuning on a matrix initialized by principal components to a compressed model from PiSSA. The key difference is that the low-rank matrix is obtained through a well-designed algorithm, while PiSSA got the matrix from a single SVD directly. Moreover, ELSA removes the mask operator in previous methods of fine-tuning compressed models and can accelerate training and convergence.

### 4.3 Combining Low-Rank Approximation and Other Model Compression Methods

There is significant research interest in merging low-rank decomposition with other compression techniques. In network quantization, recent studies like LQ-LoRA (Guo et al., 2023) and SVDQuant (Li et al., 2024) have incorporated the quantized error into a low-rank matrix. In network pruning, efforts like LoSparse (Li et al., 2023), LoRAPrune (Zhang et al., 2023), and OATS (Zhang & Papyan, 2024) propose decomposing the weight matrix into low-rank and sparse components. It is important to note that ELSA uses a more accurate objective function (minimizing approximation error) and gets the optimization faster and better, and analyzes it theoretically and empirically. Moreover, ELSA explores the advantages of fine-tuning the low-rank branch.

## 5 Conclusion and Future Work

In this paper, we explore decomposing matrix weight into sparse and low-rank matrices to better compress Large Language Models. Our work incorporates three key innovations: (1) a novel decomposition algorithm supported by theoretical proof and empirical experiments; (2) efficiently recovering the performance by fine-tuning the low-rank matrices; (3) layer-wise compression rate. Our method can better compress LLMs from different families and scales, also showing efficiency in compression and inference. One of the limitations of our method is that the structured compression form is unable to achieve considerable inference speedup. We leave it as future work to better accelerate LLMs. We hope this work can enhance the understanding of LLM compression and that the idea of low-rank and sparse approximation and adaptation can help improve the efficiency of neural networks.

## 6 REPRODUCIBILITY STATEMENT

We provide all the necessary details to reproduce our experiments in subsection 3.1 and Appendix A.1 and A.2.

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

# A APPENDIX

This supplementary material includes the following sections:

- In A.1, we provide more implementation details.
- In A.2, we provide detailed hyper-parameter for different models.
- In A.3, we provide detailed zero-shot task performance.
- In A.4, we provide approximation error curve of different layers.
- In A.5, we provide loss and gradient norm curve of recovery fine-tuning.
- In A.6, we provide ablation experiments on iteration steps.
- In A.7, we provide comparison experiment with PiSSA.
- In A.8, we describe how we use LLM in our paper.

## A.1 IMPLEMENTATION DETAILS

We provide implementation details in Table 6. Compared with LoRA finetuning, we use a smaller leaning rate and dropout rate, which is mainly because ELSA start with a smaller loss and larger gradient norm. For the matrix decomposition in low-rank and sparse approximation, we use torch.linalg.eigh() to perform eigenvalue decomposition and torch.svd_lowrank() to perform singular value decomposition.

Table 6: Implement Details

| Configuration | LoRA | ELSA |
|---|---|---|
| Optimizer name | AdamW | |
| Optimizer $\beta_1$ | 0.9 | |
| Optimizer $\beta_2$ | 0.999 | |
| Optimizer $\epsilon$ | 1e-8 | |
| Learning rate | 1e-4 | 2e-5 |
| Learning rate scheduler | cosine | |
| Batch size | 4 | |
| Training steps | 450 | |
| Warm-up steps | 22 | |
| LoRA rank | 32 | |
| LoRA alpha | 32 | |
| LoRA dropout | 0.1 | 0.05 |
| Numerical precision | bfloat16 | |

## A.2 HYPER-PARAMETERS

In this section, we provide detailed hyper-parameters for different models.

Table 7: Hyper-parameters

| Model | Rank Ratio | Iters | Layer-wise rate coefficient $\epsilon$ |
|---|---|---|---|
| LLaMA-2-7B | 0.1 | | |
| LLaMA-2-13B | 0.05 | | |
| LLaMA-3-8B | 0.2 | | |
| LLaMA-3.2-1B | 0.1 | 80 | 0.2 |
| LLaMA-3.2-3B | 0.1 | | |
| OPT-6.7B | 0.1 | | |
| Qwen2.5-7B | 0.1 | | |

## A.3 DETAILED RESULT OF ZERO-SHOT TASKS

In this section, we provide detailed zero-shot task accuracy for the experiments in main text. The results are shown in Table 8 and Table 9.

Table 8: Detailed Zero-shot Accuracy of 50% unstructured compression without finetuning

| Model | Method | Arc-c | Arc-e | BoolQ | HellaSwag | OpenbookQA | PIQA | WinoGrande | Avg. |
|-------|--------|-------|-------|-------|-----------|------------|------|------------|------|
| LLaMA-2-7B | Wanda | 0.4249 | **0.7226** | 0.7642 | 0.7098 | 0.4280 | 0.7682 | 0.6882 | 0.6437 |
| | SparseGPT | 0.4258 | 0.6877 | **0.7697** | 0.7131 | 0.4120 | 0.7682 | **0.6914** | 0.6383 |
| | ELSA | **0.4352** | 0.7189 | 0.7648 | **0.7170** | **0.4380** | 0.7699 | 0.6882 | **0.6474** |
| LLaMA-3-8B | Wanda | 0.4360 | 0.6675 | 0.7771 | 0.6935 | 0.3940 | 0.7661 | 0.7009 | 0.6336 |
| | SparseGPT | 0.4420 | 0.6873 | **0.8080** | **0.7282** | 0.4140 | 0.7709 | 0.7182 | 0.6527 |
| | ELSA | **0.4684** | **0.7252** | 0.8009 | 0.7257 | **0.4180** | **0.7851** | **0.7238** | **0.6639** |
| OPT-6.7B | Wanda | 0.3200 | **0.5829** | 0.6382 | 0.6139 | 0.3640 | 0.7486 | 0.6314 | 0.5570 |
| | SparseGPT | 0.3268 | 0.5804 | 0.6615 | **0.6296** | **0.3840** | **0.7530** | 0.6456 | **0.5687** |
| | ELSA | **0.3328** | 0.5753 | **0.6645** | 0.6104 | 0.3780 | 0.7421 | **0.6685** | 0.5674 |
| Qwen2.5-7B | Wanda | 0.4497 | 0.7239 | 0.8303 | 0.7225 | 0.4140 | 0.7818 | **0.7143** | 0.6623 |
| | SparseGPT | 0.4608 | 0.7378 | 0.8376 | 0.7363 | 0.4300 | 0.7813 | 0.7127 | 0.6709 |
| | ELSA | **0.4957** | **0.7710** | **0.8425** | **0.7380** | **0.4340** | **0.7894** | 0.7072 | **0.6826** |
| LLaMA-3.2-1B | Wanda | 0.2807 | 0.4705 | 0.6119 | 0.4420 | 0.2900 | 0.6447 | 0.5193 | 0.4656 |
| | SparseGPT | **0.2892** | 0.5025 | **0.6263** | 0.5042 | **0.3100** | **0.6801** | 0.5430 | 0.4936 |
| | ELSA | 0.2747 | **0.5253** | 0.6232 | **0.5061** | 0.3040 | 0.6746 | **0.5643** | **0.4960** |
| LLaMA-3.2-3B | Wanda | 0.3541 | 0.5850 | 0.6853 | 0.6106 | 0.3560 | 0.5364 | 0.6275 | 0.5364 |
| | SparseGPT | 0.3729 | 0.6216 | 0.7287 | **0.6440** | **0.3780** | 0.5658 | **0.6496** | 0.5658 |
| | ELSA | **0.3882** | **0.6620** | **0.7312** | 0.6437 | 0.3600 | **0.5724** | 0.6488 | **0.5724** |
| LLaMA-2-13B | Wanda | 0.4565 | 0.7151 | 0.8104 | **0.7617** | 0.4360 | 0.7922 | 0.7135 | 0.6693 |
| | SparseGPT | 0.4608 | 0.7100 | **0.8208** | 0.7571 | **0.4420** | **0.7938** | **0.7182** | 0.6718 |
| | ELSA | **0.4778** | **0.7260** | 0.8107 | 0.7577 | 0.4400 | 0.7911 | 0.7151 | **0.6741** |

Table 9: Detailed Zero-shot Accuracy of 50% unstructured compression with finetuning

| Model | Method | Arc-c | Arc-e | BoolQ | HellaSwag | OpenbookQA | PIQA | WinoGrande | Avg. |
|-------|--------|-------|-------|-------|-----------|------------|------|------------|------|
| LLaMA-2-7B | Wanda | 0.4232 | 0.6827 | **0.7514** | 0.7294 | 0.4220 | 0.7764 | 0.6764 | 0.6374 |
| | SparseGPT | 0.4181 | 0.6351 | 0.7492 | 0.7293 | **0.4360** | 0.7709 | 0.6803 | 0.6313 |
| | ELSA | **0.4497** | **0.7146** | 0.7465 | **0.7369** | 0.4340 | **0.7856** | **0.6898** | **0.6510** |
| LLaMA-3-8B | Wanda | 0.4718 | 0.7290 | 0.7813 | 0.7113 | 0.4080 | 0.7677 | 0.7182 | 0.6553 |
| | SparseGPT | 0.4974 | 0.7433 | 0.8171 | **0.7587** | 0.4280 | **0.7938** | **0.7324** | 0.6815 |
| | ELSA | **0.5145** | **0.7576** | **0.8330** | 0.7522 | **0.4440** | 0.7911 | 0.7024 | **0.6850** |
| OPT-6.7B | Wanda | 0.3259 | 0.5560 | 0.6700 | 0.6516 | 0.3900 | 0.7552 | 0.6369 | 0.5694 |
| | SparseGPT | 0.3387 | 0.5551 | 0.6657 | 0.6542 | **0.4040** | 0.7563 | **0.6472** | 0.5745 |
| | ELSA | **0.3473** | **0.5795** | **0.6807** | **0.6545** | 0.3940 | 0.7557 | 0.6433 | **0.5793** |
| Qwen2.5-7B | Wanda | **0.5418** | 0.7854 | 0.8471 | 0.7568 | **0.4580** | 0.7889 | 0.7043 | **0.6889** |
| | SparseGPT | 0.4881 | 0.7365 | **0.8529** | 0.7600 | 0.4400 | 0.7889 | **0.7088** | 0.6822 |
| | ELSA | 0.5060 | **0.7963** | 0.8330 | **0.7613** | 0.4300 | **0.7894** | 0.6977 | 0.6877 |
| LLaMA-3.2-1B | Wanda | 0.3217 | **0.5585** | **0.6346** | 0.5138 | 0.3320 | 0.6817 | 0.5485 | 0.5130 |
| | SparseGPT | **0.3225** | 0.5480 | 0.5685 | **0.5424** | **0.3440** | 0.7046 | 0.5620 | 0.5131 |
| | ELSA | 0.3208 | 0.5484 | 0.6196 | 0.5334 | 0.3420 | **0.7138** | **0.5651** | **0.5204** |
| LLaMA-3.2-3B | Wanda | 0.4104 | 0.6675 | 0.6856 | 0.6605 | **0.4000** | 0.5799 | 0.6551 | 0.5799 |
| | SparseGPT | 0.4104 | 0.6902 | 0.6917 | 0.6802 | 0.3980 | 0.5887 | 0.6614 | 0.5887 |
| | ELSA | **0.4309** | **0.6734** | **0.7055** | **0.6827** | 0.3880 | **0.5905** | **0.6622** | **0.5905** |
| LLaMA-2-13B | Wanda | 0.4667 | 0.6911 | 0.7976 | 0.7741 | **0.4640** | 0.8030 | 0.7096 | 0.6723 |
| | SparseGPT | 0.4778 | 0.6907 | 0.8092 | **0.7758** | 0.4500 | 0.8009 | **0.7127** | 0.6739 |
| | ELSA | **0.4821** | **0.6932** | **0.8113** | 0.7720 | 0.4580 | **0.8036** | 0.7103 | **0.6758** |

## A.4 APPROXIMATION ERROR CURVE

In this section, we provide the detailed approximation error curve of different methods in Figure 4 and Figure 5. We choose 1st, 16th, 32nd layers in LLaMA-2-7B for this experiment. The results shows that our methods has the smallest approximation error except for the q_proj and k_proj in the first layer. We find that only in a small part of layers, the approximation error of the decomposition is not the smallest. Replacing with best decomposition method will cause little difference. Therefore, we adopt low-rank and sparse decomposition for all layers.

These results also show that most layers in LLM are closer to a sparse matrix than a low-rank matrix. It's noticed that the error of sparse only is smaller than low-rank matrix. This implies that most layers in LLMs are closer to sparse matrix than low-rank matrix. But this may not imply that the low-rank component is merely a minor correction to a fundamentally sparse weight matrix. If we run only 1 iteration, which means do a pruning first and run truncated SVD on the residual matrix, the

result is worse than pruning in many cases. The initial point of OATS curve represents this case. In many matrices, the error is larger than pruning. This implies the low-rank component is not a minor correction.

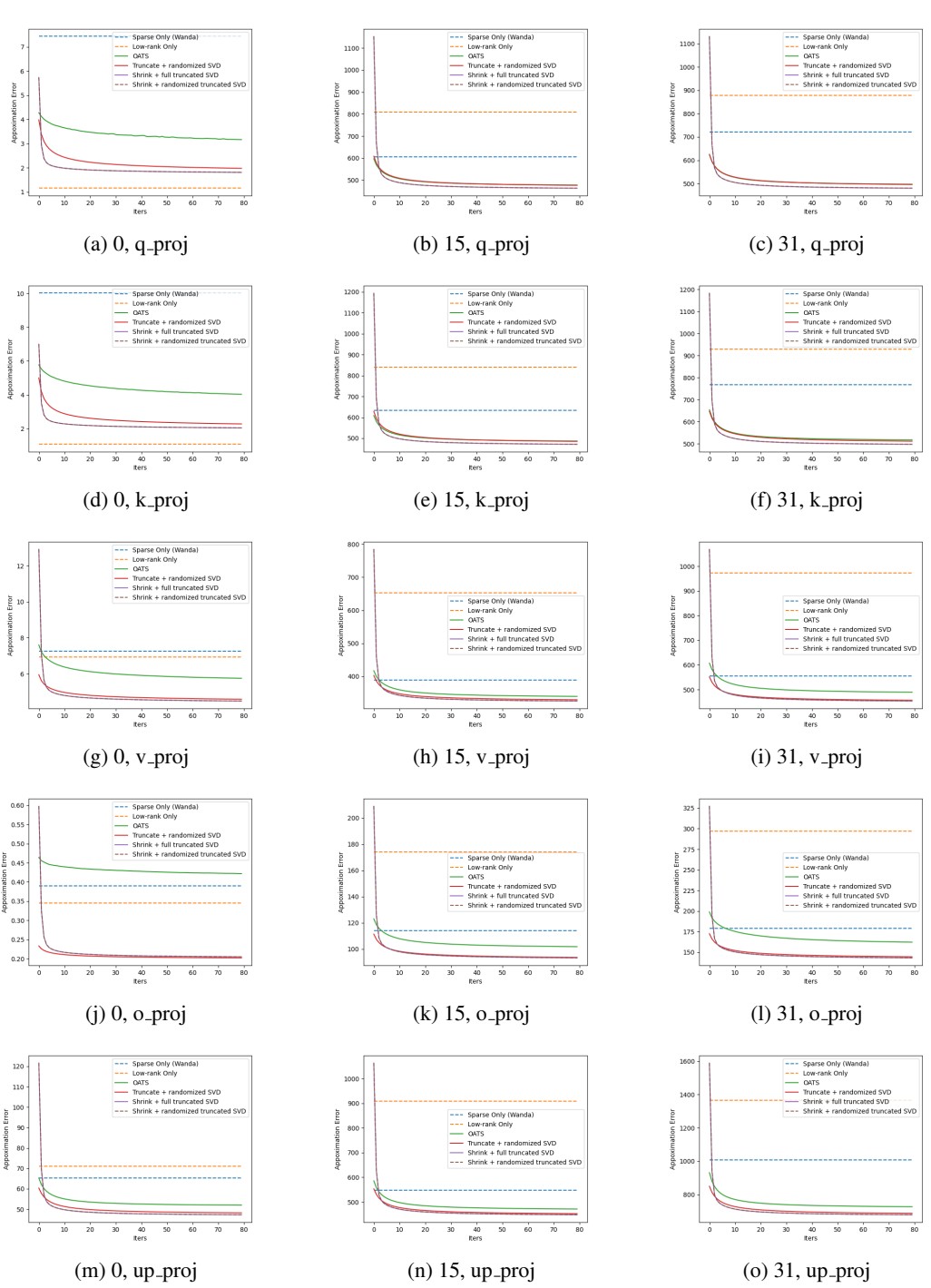

Figure 4: Approximation Error Curve

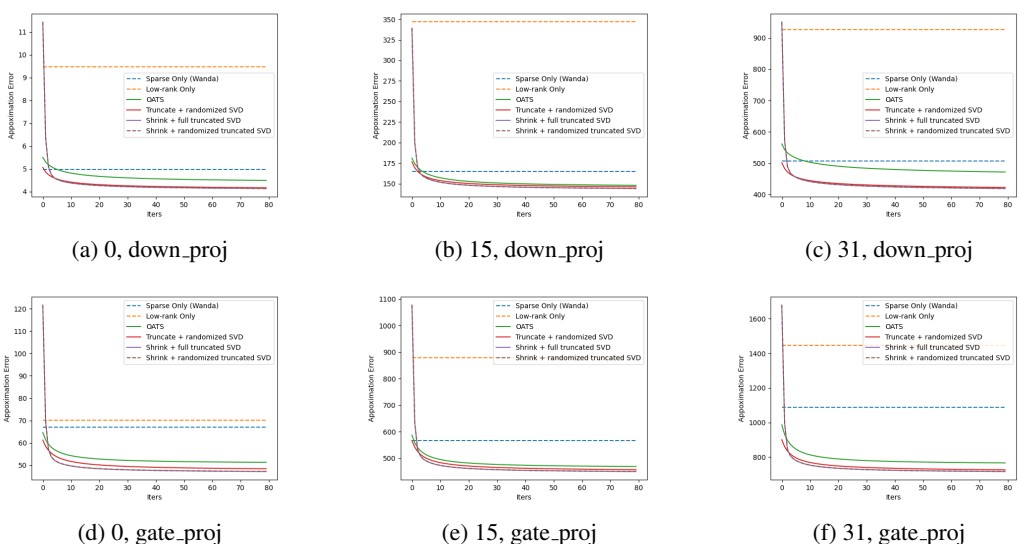

(a) 0, down_proj     (b) 15, down_proj     (c) 31, down_proj

(d) 0, gate_proj     (e) 15, gate_proj     (f) 31, gate_proj

Figure 5: Approximation Error Curve (continue)

## A.5 TRAINING CURVE

Figure 6a, 6b show the loss curve and gradient norm curve of different finetuning strategies. The loss curve shows that ELSA starts with a smaller loss, and the gradient norm shows that it has a larger gradient and converges faster.

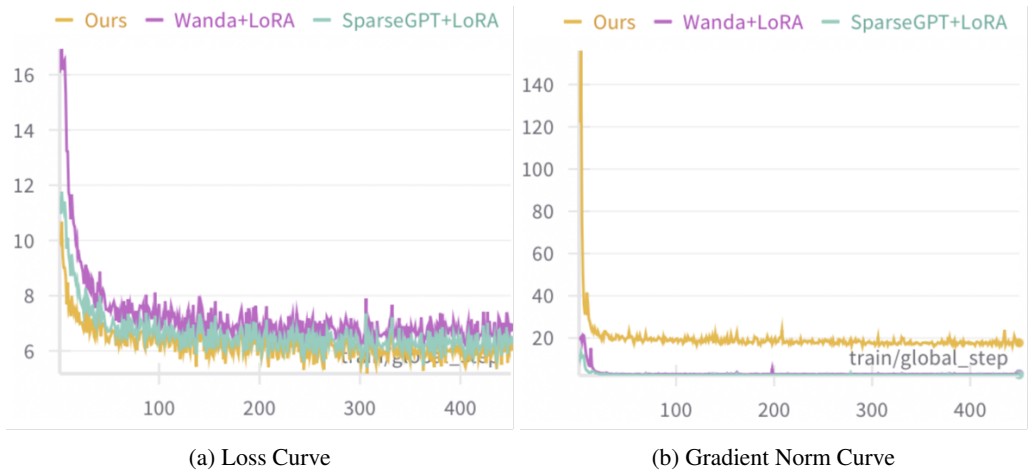

(a) Loss Curve          (b) Gradient Norm Curve

Figure 6: Training curve of LLaMA-2-7B.

## A.6 ABLATION EXPERIMENTS ON ITERATION STEP

We run ELSA across different iteration steps on LLaMA-2-7B under 50% unstructured setting. Table 10 shows that the performance is better as the iteration step increases. And we set the step to 80 for the best performance.

## A.7 COMPARISON AGAINST PISSA

ELSA extends the idea of finetuning on a matrix initialized by principle components to compressed model from PiSSA. The key difference is that the low-rank matrix is obtained through a well-designed

Table 10: Ablation on Iteration step

| Iters | 20 | 40 | 60 | 80 | 100 |
|---|---|---|---|---|---|
| PPL($\downarrow$) | 6.73 | 6.68 | 6.64 | **6.57** | 6.61 |
| 0-shot acc($\uparrow$) | 64.27 | 64.47 | 64.72 | **64.74** | 64.54 |

algorithm, while PiSSA got the matrix from a single SVD directly. We conduct an experiment to test if directly applying PiSSA (pruning with Wanda/SparseGPT and initializing the matrix as the principle component of $W - S$) works. The result on LLaMA-2-7B under 50% unstructured results are shown in Table 11. This experiment shows the effectiveness of the decomposition method against PiSSA.

Table 11: Comparison against PiSSA

| | Wanda + PiSSA init | SparseGPT + PiSSA init | ELSA |
|---|---|---|---|
| PPL ($\downarrow$) | 6.65 | 6.69 | **6.49** |
| 0-shot acc ($\uparrow$) | 64.07 | 64.21 | **65.10** |

## A.8 THE USE OF LARGE LANGUAGE MODEL (LLM)

In this paper, we employed LLM to check grammar during the writing process.

