# OpenReview forum: "Efficient Low-rank and Sparse Approximation and Adaptation for Large Language Models"
_ICLR.cc/2026/Conference — ICLR 2026 Conference Withdrawn Submission_

### Official Review · Reviewer_xNpa · 2025-10-24

**Soundness:** 2
**Presentation:** 2
**Contribution:** 1
**Rating:** 2
**Confidence:** 5

**Summary:**

This paper studies the sparse + low rank compression problem for LLMs. They propose an alternating optimization framework. The sparse part is solved using a drop-in pruning method (Wanda + modifications). The low-rank part is solved in closed form. They discuss structured and unstructured sparsity. Numerical experiments are presented with comparisons with structured and unstructured pruning baselines.

**Strengths:**

- The paper studies an important problem.
- The inference speed experiments are welcome.

**Weaknesses:**

- Writing: Although I don't think the paper is poorly written, there are some issues with writing. For example:

Comparison with Lora: Figure 1 is confusing. The difference with Lora is that the low-rank part computation is overlapped with the sparse part, instead of being merged into a single weight? This is an inference implementation difference which is not even a major focus of this paper. From the algorithmic perspective, what this paper discusses is an initialization for Lora (as stated by the authors). Additionally, authors claim that *There are existing techniques for
speeding up sparse matrix multiplications. By employing these techniques, frozen sparse matrices
can boost the speed of LoRA finetuning*. This is an extreme simplification when it comes to GPUs. There is no strictly monotonic relationship between sparsity and matrix multiplication speed on GPUs. There is no widely adopted framework that makes training with sparse matrices faster on GPUs. This statement should be ideally removed as it is vague and misleading.

- Baselines: The paper is missing the baseline https://proceedings.mlr.press/v280/makni25a.html. In fact, the same paper discusses using drop-in pruning methods for the sparse subproblem.

In addition, there are newer pruning baselines, such as https://arxiv.org/abs/2406.07831 and https://arxiv.org/abs/2310.08915 which should be included in the experiments.

Moreover, I'm not sure I follow what is happening with the structured pruning part. Are you making each layer's weights sparse? In other words, what does 20% structured mean in Table 2? Structured pruning methods choose specific layers to sparsify, based on how that would benefit inference. For example, see https://arxiv.org/abs/2403.12983. Please update the experiments to follow proper structures and please add the baseline mentioned here.

**Questions:**

- Why substitute truncation with soft thresholding in Wanda? There needs to be more ablation studies (I believe there is only Table 5 regarding this question which is limited).

---

### Official Review · Reviewer_ngHE · 2025-10-31

**Soundness:** 2
**Presentation:** 2
**Contribution:** 1
**Rating:** 2
**Confidence:** 4

**Summary:**

This paper proposes a method named ELSA for the compression and adaptation of Large Language Models. By decomposing weight matrices into a combination of sparse matrices and low-rank matrices, ELSA achieves efficient model compression while supporting efficient recovery fine-tuning. The method comprises an alternating projection algorithm, a LoRA-compatible fine-tuning strategy, and layer-wise compression rate allocation. Experiments validate its superiority across multiple LLM families (e.g., LLaMA, OPT, Qwen) and tasks, demonstrating improvements in performance, compression efficiency, and inference speed.

**Strengths:**

1. The paper is well-written and highly readable, with a clear structure and detailed pseudocode for the algorithm.
2. Pruning techniques generally suffer from performance degradation and require post-pruning fine-tuning to compensate for performance loss, and ELSA provides a practical solution to this problem.
3. Extensive experiments are conducted in the paper across different model families (LLaMA, OPT, Qwen), scales (1B to 13B parameters), and compression settings (50% unstructured, 2:4 semi-structured, 20% structured). These experiments include zero-shot tasks, perplexity evaluations, and hardware acceleration tests (CPU/GPU).

**Weaknesses:**

1. The novelty is limited. Decomposing the weight matrix into sparse and low-rank components (W ≈ L + S) is not an entirely new concept. In the field of model compression, several recent works have adopted similar ideas, such as OATS [1], LoSparse [2], and LoRAPrune [3]. The method proposed in this paper is highly similar to these works in terms of its framework.
2. The main contribution of the paper seems to be a new combination of these existing technical modules. For instance, the layer-wise compression rate allocation adopts the methods proposed in ShortGPT [4] and OWL [5], which is more like an engineering improvement rather than the proposal of a fundamentally new algorithm.
3. The performance improvement is quite limited. For example, in the case of 50% unstructured pruning on the LLaMA2-13B model (Table 2), the average zero-shot accuracy only increases by 0.17% after fine-tuning, and the perplexity only decreases by 0.22. Additionally, the performance improvement of fine-tuning the structurally pruned LLaMA2-13B model is also quite limited. Therefore, the ELSA algorithm does not seem to be effective.
4. The performance contribution of different components is rather limited. After adding SHR, and LWC , the average zero-shot accuracy only increases by 0.66%, and the perplexity only decreases by 0.19 (Table 5)—these are extremely minor improvements. Thus, the proposed components do not seem to be necessary.

[1] Zhang S, Papyan V. OATS: Outlier-Aware Pruning Through Sparse and Low Rank Decomposition [C]. The Thirteenth International Conference on Learning Representations.

[2] Li Y, Yu Y, Zhang Q, et al. Losparse: Structured compression of large language models based on low-rank and sparse approximation [C]. International Conference on Machine Learning. PMLR, 2023: 20336-20350.

[3] Zhang M, Chen H, Shen C, et al. LoRAPrune: Structured Pruning Meets Low-Rank Parameter-Efficient Fine-Tuning [C]. ACL (Findings). 2024.

[4] Men X, Xu M, Zhang Q, et al. Shortgpt: Layers in large language models are more redundant than you expect [J]. arXiv preprint arXiv:2403.03853, 2024.

[5] Yin L, Wu Y, Zhang Z, et al. Outlier Weighed Layerwise Sparsity (OWL): A Missing Secret Sauce for Pruning LLMs to High Sparsity [C]. International Conference on Machine Learning. PMLR, 2024: 57101-57115.

**Questions:**

1. How does τ in the Shrink function affect the final performance? Can more ablation experiments be provided?
2. How does ELSA perform at other sparsity rates? For instance, in the case of 60% and 70% unstructured pruning.

---

### Official Review · Reviewer_DcGJ · 2025-10-31

**Soundness:** 2
**Presentation:** 1
**Contribution:** 3
**Rating:** 4
**Confidence:** 2

**Summary:**

This paper introduces ELSA that combines pruning and low rank approximation for better performance with training. The weight matrices will be first decomposed into sparse matrices and low-rank matrices using alternative initilaization. To better recovery the performance, the author freezes the sparse matrices and updates the low-rank matrices. Emperical experiments on different models and benchmarks show this method can outperform baselines and improve the inference speed and compressing speed with different compression ratios.

**Strengths:**

- This paper tackles important problem of model compression with methods inspired from widely used algrorithms.
- This paper compares with different baselines and achieve better performance on both the performance and the efficiencies

**Weaknesses:**

- The writing is quite confusing to me. I can understand the main idea of this paper, which use alternative optimization to init the sparse and low rank components and then update it using LoRA. Authors present a lot of details of how this is done, but provide minor motivation figures about whether indeed the optmization goal in 2.1 is satisfied.
- To recovery the performance, only the low-rank part is updated. The author mentions in the paper about this choice in line 233 and 234 that for not destorying the sparsity. I think we can still train the sparse matrix we want in many different ways. There are many papers about dynamic pruning during training. It's very strange for me why not collectively optimize the sparse part and the low-rank part collectively.\
- In the paper writing part, lack of the exploration of this idea with previous ideas. Sparse plus low rank approximations is not a novel idea. I didn't see enough discussions of why this paper is novel and unique compared to previous papers.

**Questions:**

In proof theorem 2.1, even when fixing the S, maybe define a W'=W-S would be more appropriate?

Though authors introduce the optimization goal in 2.1, training is still employed to recover the performance. I wonder won't training break the optmization goal as introduced in 2.1?

---

### Note · Authors · 2025-11-28

I have read and agree with the venue's withdrawal policy on behalf of myself and my co-authors.